# A Wideband Folded Dipole Antenna with an Improved Cross-Polarization Level for Millimeter-Wave Applications

**Lianpeng Xue, Qiangquan Tan** , **Ke Cheng and Kuikui Fan** *

School of Electronics and Information, Hangzhou Dianzi University, Hangzhou 310018, China
* Correspondence: kkfan@hdu.edu.cn

**Abstract:** A low-profile planar millimeter-wave (MMW) folded dipole antenna fed by substrate integrated waveguide (SIW) is proposed in this letter. By etching the gaps at the proper position of 1.5λ dipole, an additional resonant mode is generated. Accordingly, the working bandwidth is greatly broadened. In addition, by appropriately adjusting the length of the dual-side parallel strip line (DSPSL), the radiated electric fields generated by the aperture of the feeding SIW and the connecting metallic vias of the folded dipole are designed with an out-of-phase potential. Hence, the cross-polarization of the presented folded dipole antenna is improved as well. As a demonstration, a prototype is fabricated and measured. The experimental results exhibit that the proposed folded dipole has a −10 dB impedance bandwidth of 58.5% (from 30.3 GHz to 53.7 GHz), a gain of around 5 dBi with more than 120 degrees beamwidth in H-plane, and a cross-polarization levels below −15 dB, covering the working frequency band. Compared with the up-to-date planar dipole antenna, the proposed folded dipole achieves the widest working bandwidth and low cross-polarization level. The proposed antenna can be used as the terminal antenna of the MMW communication system.

**Keywords:** substrate integrated waveguide (SIW); the folded dipole; wideband antenna; millimeter-wave antenna

## 1. Introduction

Due to the global bandwidth shortage and the rapid increase of mobile data growth, millimeter-wave band is considered as a candidate spectrum for the fifth generation (5G) mobile communication network. As an important component, printed dipole antennas were considered to be highly potentially useful for their attractive merits such as compact size, low fabrication cost, and suitability for integration with the feeding network.

Many balanced feeding networks have been proposed to feed the printed dipole antenna. Typical feeding methods such as the microstrip feeding technique [1,2] or the coplanar waveguide (CPW) feeding technique [3,4] require a balun to realize the mode transformation, which not only enlarges the antenna volume but also impairs its radiation performance. To overcome those drawbacks, the substrate integrated waveguide (SIW), which has merits of low cost, planar form, compact size, and high-density integration [5,6], has been adopted as a wideband balun to feed the printed dipole antenna at MMW band [7–11] due to the fact that the phase difference between the current along the top and bottom layers is intrinsically 180 degrees. It can therefore significantly reduce the design complexity and the losses of MMW circuits.

To expand the operating bandwidth of the MMW planar dipole antenna, a number of techniques have been reported in recent years, including increasing the number of dipoles based on the log-periodic structure [7], inserting third-order inductive windows into the SIW [8], using a wideband integrated balun structure [12], and employing novel ladder-like directors [13]. Compared to the previously mentioned methods, the method of using the folded structure, which does not result in the increase of the area and complexity of the antenna, is a more effective way of expanding the operating bandwidth of the dipole

antenna [13–15]. In previous work [16], a broadband SIW-fed dipole antenna has been developed by using long printed arms with two gaps. It can work at the half-wavelength and three half-wavelength dipole modes. However, the cross-polarization of the wideband dipole antenna is poor above −13 dB due to the vertical electric field generated by the aperture of the SIW terminal.

In this paper, a broadband SIW-fed folded dipole with improvement of cross-polarization is presented. Two gaps are cut at the 1.5λ printed long dipole to generate additional resonance, which, combined with the existing resonance, are utilized to enhance the working bandwidth. The SIW is adopted to feed the folded dipole rather than the complex balun structure, and a dual-side parallel strip line (DSPSL) with a characteristic impedance of around 90 ohm is directly connected to the folded dipole element and SIW, without demanding taper transitions [17,18]. Compared with the previously reported wideband dipole antennas, the presented folded dipole antenna shows a considerable increase in the impedance bandwidth while keeping a compact structure, as well as an improvement in cross-polarization level, which is around 9 dB lower than that of the conventional SIW-fed dipole antenna.

## 2. Materials and Methods

### 2.1. Configuration of the Folded Dipole

Figure 1 depicts the configuration of the proposed folded dipole element with optimized sizes, as shown in Figure 1. It consists of a DSPSL and a folded long dipole with two gaps. A SIW is used to feed the antenna as a broadband balun. The folded long dipole is composed of two identical rectangular strips printed on the bottom and top surfaces of the substrate, respectively. The rectangular strips are connected by shorted metal vias. By etching the gaps at the zero current points, the folded long dipole can work at half-wavelength mode and three half-wavelength mode. Thus, the working bandwidth can be broadened. The metal vias is used to improve cross polarization.

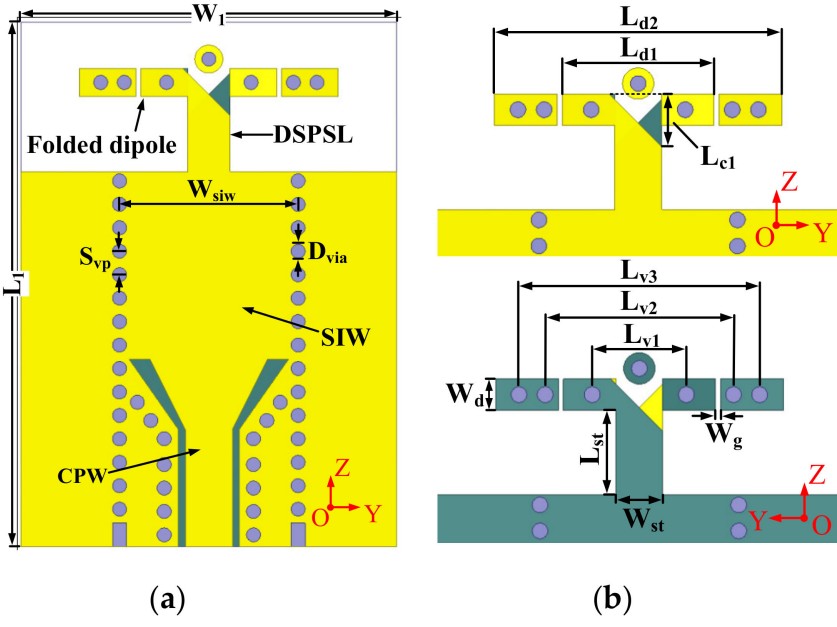

**Figure 1.** (**a**) Configuration of the proposed SIW-fed folded dipole antenna (**b**) top view and bottom view of the folded dipole.

The proposed antenna is designed on RT/Duriod 5880 substrate with the thickness of 0.508 mm and its permittivity and loss tangent are 2.2 and 0.0009, respectively. The center frequency of the proposed antenna is 34 GHz. The proposed antenna was optimized to

achieve a high gain and a wideband operating bandwidth by using the ANSYS HFSS. The optimized parameters for the final design are given in Table 1.

**Table 1.** The dimensions of the optimized folded dipole antenna. (Unit: mm).

| $W_1$ | $L_1$ | $W_{siw}$ | $S_{vp}$ | $D_{via}$ | $W_{st}$ | $L_{st}$ | $L_{d1}$ |
|-------|-------|-----------|----------|-----------|----------|----------|----------|
| 8 | 14 | 3.8 | 0.5 | 0.3 | 0.9 | 1.6 | 2.9 |
| $L_{d2}$ | $W_d$ | $W_g$ | $L_{c1}$ | $L_{v1}$ | $L_{v2}$ | $L_{v3}$ | |
| 5.5 | 0.6 | 0.1 | 1.2 | 1.8 | 3.6 | 4.6 | |

### 2.2. The Principle of the Dual Mode Operation

Compared with the conventional SIW-fed dipole antenna [8–10], the proposed folded dipole antenna achieves a wide operating bandwidth owing to dual mode resonance. For a dipole antenna whose diameter is much smaller than the wavelength, current flowing in the arms of dipole has a sinusoidal standing wave pattern with a null at the end. Figure 2 shows the current distribution corresponding to dipoles with different arm lengths. For $0.5\lambda$ dipole, the currents have the same phase and amplitude in the two arms and the input impedance of $0.5\lambda$ dipole mode is about 73 Ω. For $1.5\lambda$ dipole, there is opposite-phase currents on the arms at the same time and two current zeros located at $\pm\lambda/4$ can be observed. The input impedance of the $1.5\lambda$ dipole is calculated as 103 Ω. The input impedance of the two dipoles with different lengths is very close. Thus, if two gaps are cut at the zero current positions of $1.5\lambda$ dipole to ensure that the current distribution of $1.5\lambda$ higher-order dipole mode is not destroyed, it is possible to achieve functionality in the $0.5\lambda$ dipole mode and the $1.5\lambda$ dipole mode.

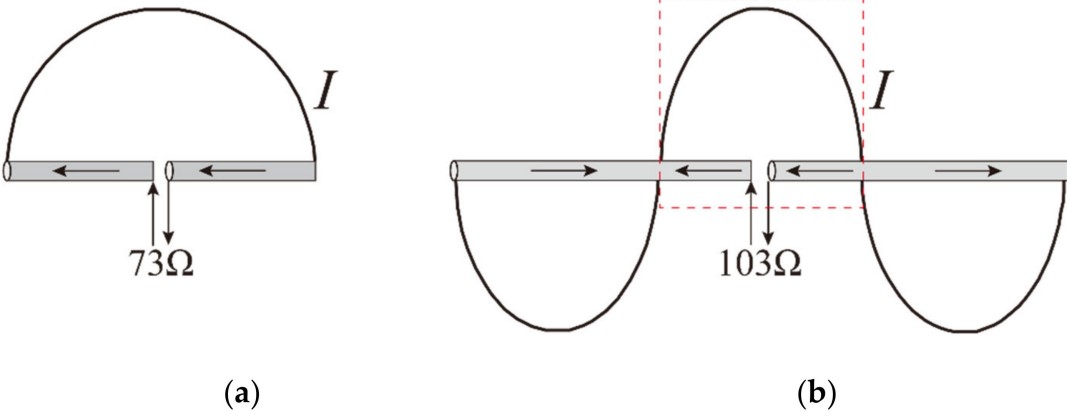

(a)                                        (b)

**Figure 2.** The current distribution of dipole with different lengths: (**a**) 0.5 λ dipole; (**b**) 1.5 λ dipole.

Figure 3 shows the evolution of wideband printed long dipole with dual resonant mode. In order to generate a half-wavelength dipole mode, the gap should be cut at zero currents to avoid destroying the original higher-order mode. By tuning the gaps to the proper position, the $1.5\lambda$ printed dipole antenna can operate with two resonant modes, i.e., a $0.5\lambda$ dipole mode and a $1.5\lambda$ dipole mode. The results have been verified in our previous work and the detailed design for the working mechanism can be seen in [16]. For the sake of brevity, the design process and the analysis of operating principles will not be illustrated here. In this work, the improvement of cross-polarization of the dual mode dipole will be elaborated.

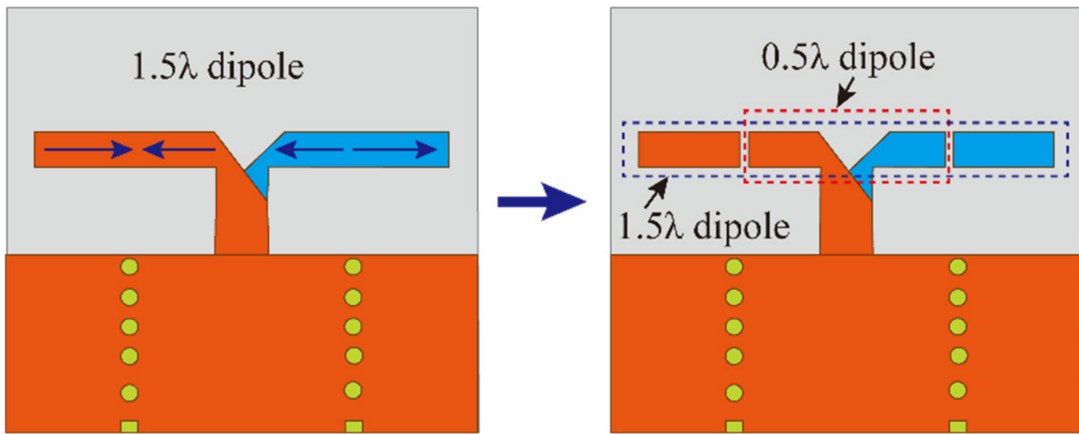

**Figure 3.** The evolution of wideband printed long dipole with dual resonant mode.

### 2.3. Improvement of Cross-Polarization

Figure 4 presents two different SIW-fed dipole antennas. Antenna A is similar to the traditional printed dipole antenna fed by SIW. Antenna B is the proposed folded dipole fed by SIW. The positions of the connecting metallic vias and the length of the arms are optimized to realize a good impedance matching. In order to reduce the parasitic effect of the SIW-to-CPW transition, both of the antennas are fed by wave ports for simulation. The simulated $|S_{11}|$ of the two investigated dipole antennas are shown in Figure 5a. The antenna works in the traditional half-wavelength dipole mode. Thus, it has only one resonance at 42 GHz and has a narrow impedance bandwidth. According to the above broadband principle, the designed SIW-fed folded dipole has two resonances, which are the traditional half-wavelength dipole mode at a low frequency and the high-order dipole mode at a high frequency. Thus, the proposed folded dipole realizes a wide impedance bandwidth of 58% covering 30 to 55 GHz. The simulated maximum gain and cross-polarization of the two antennas are shown in Figure 5b. The simulated results display that both antennas have a comparable gain that varies between 5 to 7 dBi from 30 to 55 GHz. However, the cross-polarization performance of Antenna B is greatly improved compared with that of Antenna A due to the introduction of the folded structure.

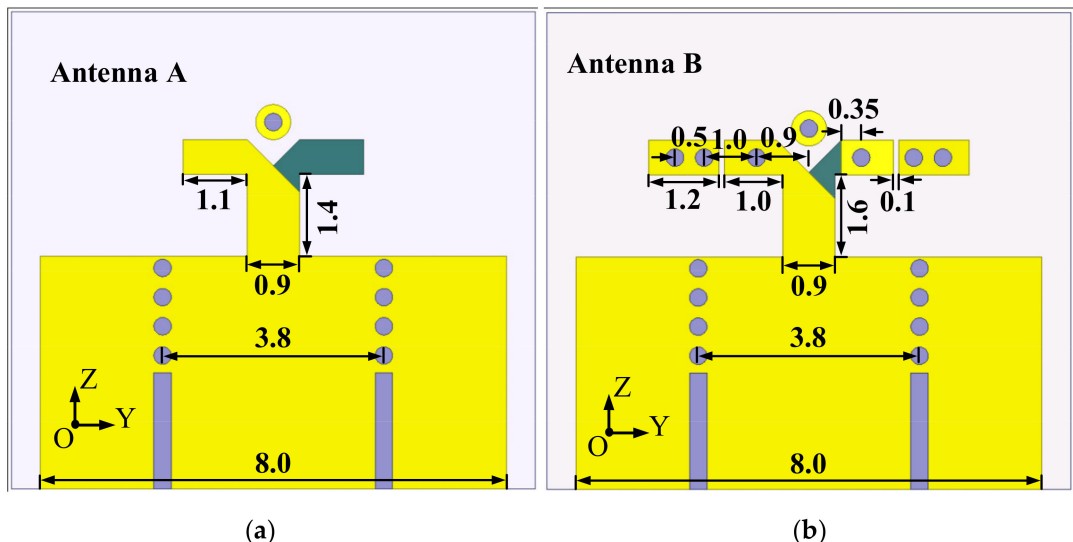

**Figure 4.** Two different SIW-fed printed dipole antennas: (**a**) routine half-wavelength dipole; (**b**) the proposed wideband folded dipole.

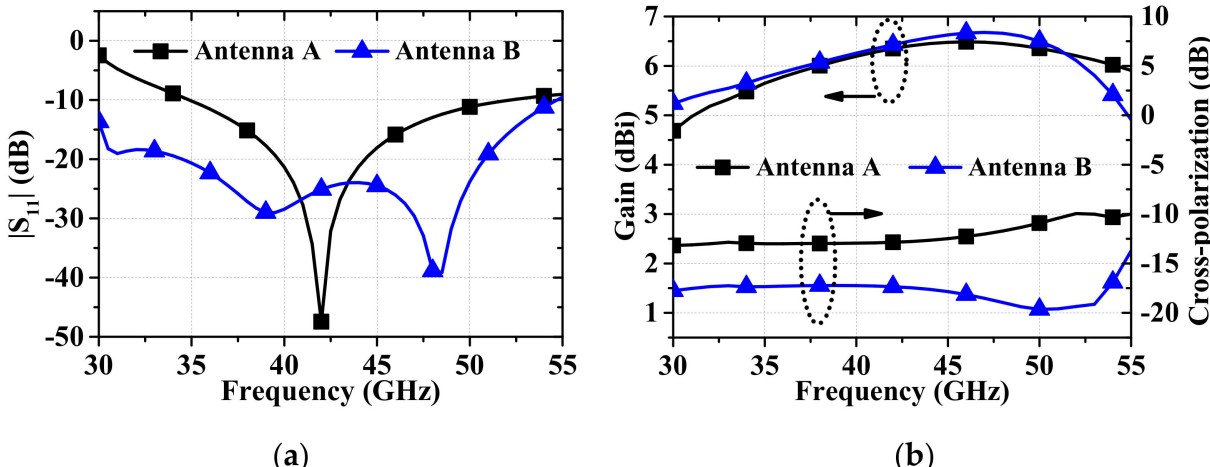

(**a**)　　　　　　　　　　　　　　　　　(**b**)

**Figure 5.** The simulated |S$_{11}$|, gain and cross-polarization of Antenna A and Antenna B: (**a**) |S$_{11}$|; (**b**) gain and cross-polarization.

Figure 6 shows the surface currents for the conventional SIW-fed dipole (Antenna A) and the presented SIW-fed folded dipole (Antenna B) at 45 GHz. The aperture of the SIW terminal is equivalent to a magnetic dipole [19] aligning with the y-axis, and it mainly contributes to the vertical polarization electric field while the printed electric dipole mainly contributes to the horizontal polarization electric field, as shown in Figure 6a. Therefore, the conventional SIW-fed dipole has a poor cross-polarization higher than −13 dB. However, for the SIW-fed folded dipole antenna, it can be observed from Figure 4b that the surface of the connecting metallic vias have a current in vertical direction and the currents have the same phase and amplitude. Therefore, the vertical polarization electric field of the SIW-fed folded dipole comes from two parts: one is the magnetic dipole and the other is the surface currents on the surface of the connecting metallic vias. If the two vertical polarization electric fields are adjusted with the out-of-phase, the cross-polarization of the SIW-fed folded dipole will be improved.

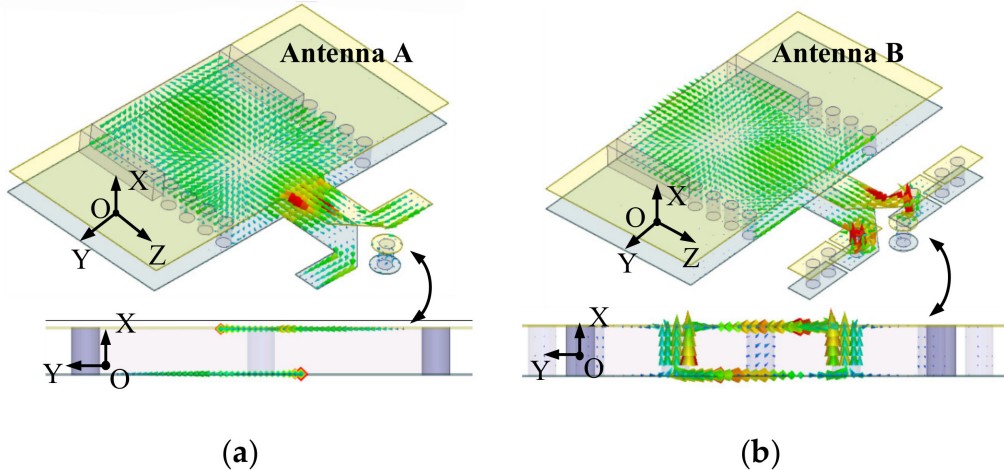

(**a**)　　　　　　　　　　　　　　　　　(**b**)

**Figure 6.** The surface electric current distribution for the two antennas at 45 GHz: (**a**) Antenna A, (**b**) Antenna B.

The relationship between the two electric fields generated by the magnetic dipole and the connecting metallic vias is studied to investigate the cross-polarization improvement. The conceptual configuration consisting of a magnetic dipole and two short electric dipoles is shown in Figure 7. The aperture serves as a virtual magnetic dipole, which is located

on the y-axis and operates at one-half-wavelength mode. Thus, the electric field of the aperture at an arbitrary observation point P in the far field is found to be [20]

$$\vec{E}_{aperture} = -\nabla \times \vec{F}_y = jkK_1\left(-\hat{\theta}\cos\varphi + \hat{\varphi}\cos\theta\sin\varphi\right)\frac{e^{-jkr}}{r} \tag{1}$$

where $k = 2\pi/\lambda$ is the wave number, and $K_1 = \mu_0 M_0 L_1/4\pi$ is the total amplitude of the vector potential function.

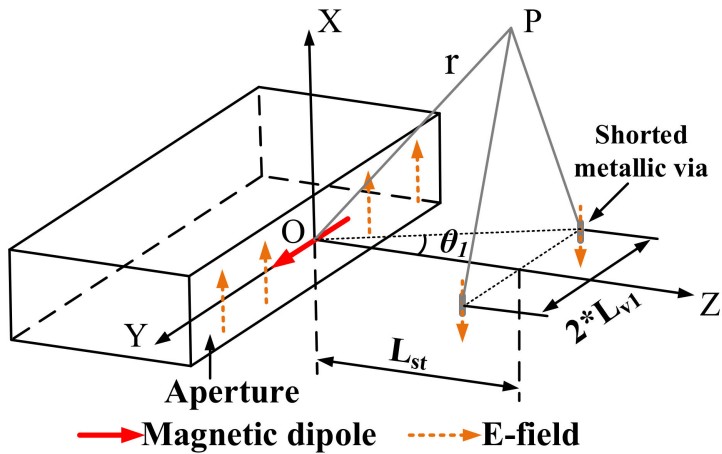

**Figure 7.** The equivalent source's model for the SIW-fed folded dipole antenna.

The connecting metallic vias are located on the two sides of the z-axis along the x-axis; the horizontal distance between the magnetic dipole and the connecting metallic vias is represented as $L_{st}$. Thus, the electric field generated by the shorted metal via at an arbitrary observation point P in the far field can be found to be

$$\vec{E}_{hole} = \frac{1}{j\omega\varepsilon\mu}\nabla \times \nabla \times \vec{A}_x = -j\frac{k^2 K_2}{\omega\mu\varepsilon}\left(\hat{\theta}\cos\theta\cos\varphi - \hat{\varphi}\sin\varphi\right)\frac{e^{-jkr}}{r} \tag{2}$$

where $k = 2\pi/\lambda$ is the wave number, and $K_2 = \mu_0 I_0 L_2/4\pi$ is the total amplitude of the vector potential function.

Then, the total electric field in far field produced by the magnetic dipole and the two shorted metal holes can be derived as

$$\vec{E}_{total} = \vec{E}_{aperture} + e^{-j(k\frac{L_{st}}{\cos\theta_1}\cos(\theta-\theta_1)+\delta_0)}\vec{E}_{hole1} + e^{-j(k\frac{L_{st}}{\cos\theta_1}\cos(\theta+\theta_1)+\delta_0)}\vec{E}_{hole2} \tag{3}$$

when $\theta = 0°$, $\varphi = 0°$, the E-field along the +z axis is

$$\begin{aligned}\vec{E}_{total} &= -\hat{\theta}\left(jkK_1 + j\frac{k^2 K_2}{\omega\mu\varepsilon}e^{-j(kL_{st}+\delta_0)}\right)\frac{e^{-jkr}}{r} \\ &= -\hat{\theta}\left(|E_1| + e^{-j(kL_{st}+\delta_0)}|E_2|\right)\frac{e^{-jkr}}{r}\end{aligned} \tag{4}$$

It is seen that the vertical electric field is mainly controlled by the length $L_{st}$ of the DSPSL. When $kL_{st} + \delta_0 = \pi$, the electric fields generated by the magnetic dipole and the connecting metallic holes are out-of-phase in the vertical direction, the amplitude of the vertical component electric field has a minimum value of $|E_1| - |E_2|$. Thus, the cross polarization of our proposed antenna has the optimal value.

The orthogonal gain components Gain($\theta$) and Gain($\varphi$) of Antenna A and the proposed Antenna B for various $L_{st}$ are shown in Figure 8a,b, respectively. For Antenna A, as the length $L_{st}$ of the DSPSL increases, the amplitude of Gain($\varphi$) will be reduced, while the amplitude of Gain($\theta$) only has a small change. A similar change can be observed for the Gain($\varphi$) of Antenna B with that of Antenna A. However, the amplitude of the Gain($\theta$) can be effectively reduced by adjusting the length of DSPSL; the proposed antenna can achieve a lower cross-polarization than the conventional SIW-fed dipole antenna.

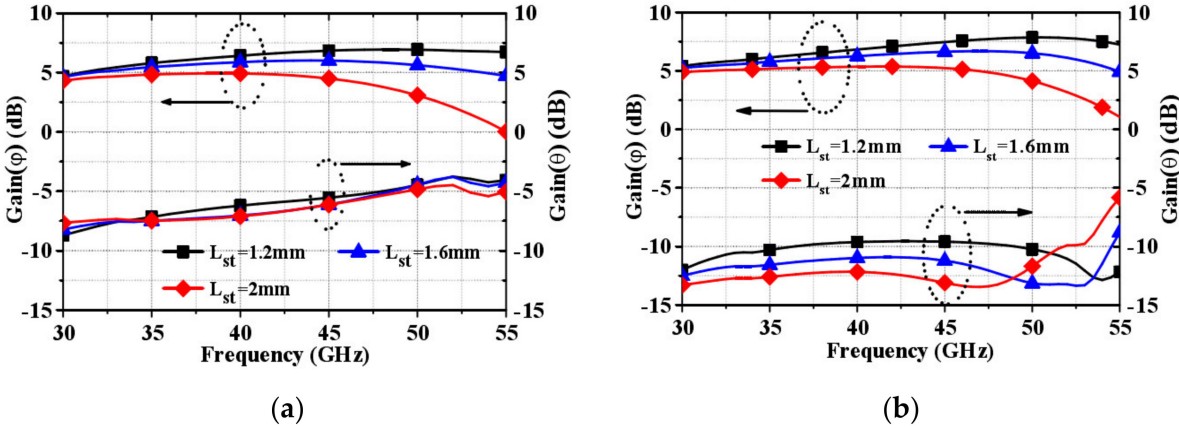

(a)    (b)

**Figure 8.** The orthogonal gain components (Gain($\theta$) and Gain($\varphi$)) of Antenna A and our proposed Antenna B; (**a**) Antenna A with different $L_{st}$. (**b**) Antenna B with different $L_{st}$.

### 3. Results

A wideband SIW-to-CPW transition is designed for measurement [21]. A prototype of the proposed antenna is fabricated using the single-layer printed circuit board (PCB) process, and the photograph is exhibited in Figure 9a. The optimized configuration is given in Table 1. The simulated and measured reflection coefficients are displayed in Figure 9b. It can be observed that the working bandwidth of proposed planar folded dipole is 55.7% (from 30.3 to 53.7 GHz), and the simulated and measured results are in good agreement. The measured gain and the cross-polarization are presented in Figure 10a and the simulated results are compared. The measured gain is between 5.1 and 6.4 dBi within the operating band. The maximum gain variation is less than 1.3 dBi. The measured cross-polarization levels are below −15 dB over the working band. It is worth noting that it is possible to make a design with a better cross-polarization at the expense of the antenna gain. Here, we make a compromise that the proposed antenna has a moderately high gain and a relatively low cross-polarization over the operating band. The simulated and the measured efficiency are also presented in Figure 10b; the measured efficiency is larger than 80%.

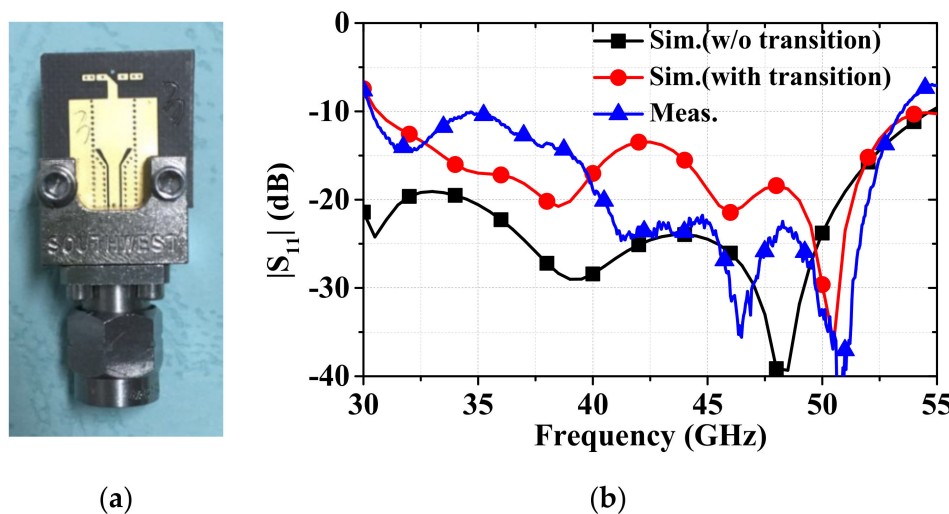

(a)    (b)

**Figure 9.** (**a**) The fabricated photograph. (**b**) The measured and simulated reflection coefficients of the proposed antenna.

The radiation patterns in the yoz-plane and the xoz-plane for 35 and 45 GHz are shown in Figure 11a,b, respectively. Stable endfire radiation patterns are obtained and the measured radiation patterns well agree with the simulated ones. However, the cross-

polarization patterns are asymmetric in the xoz-plane, which was probably caused by the SIW-to-CPW transition. The slots of the transition are distributed at one side of the substrate; it is expected to produce some leakage which make the cross-polarization patterns asymmetrical in the xoz-plane.

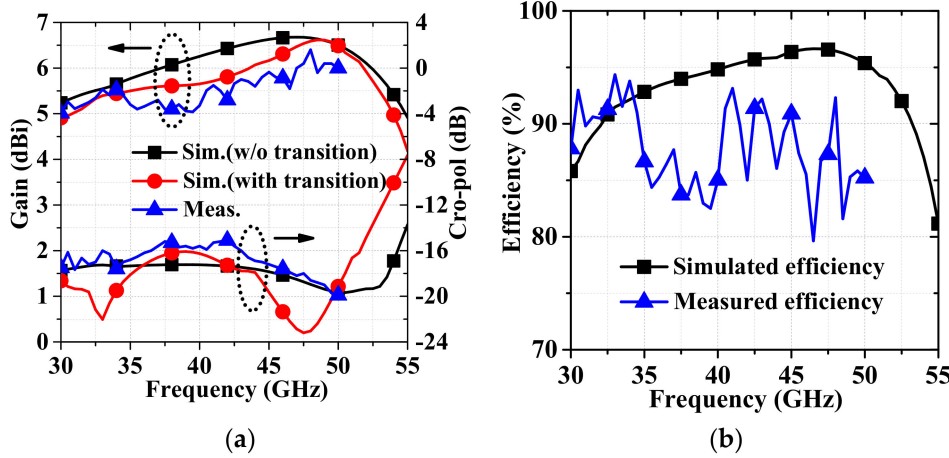

**Figure 10.** (**a**) Measured and simulated gain, cross-polarization and (**b**) efficiency of the proposed antenna.

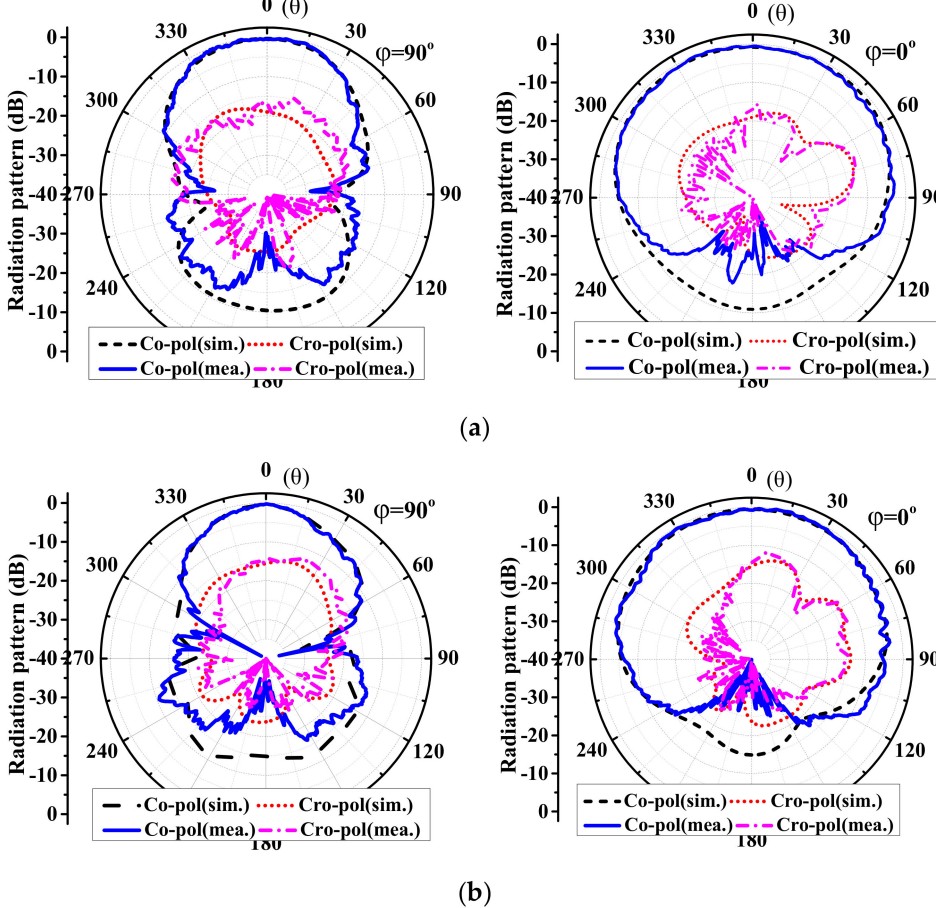

**Figure 11.** Radiation patterns for the proposed antenna in the yoz-plane (E-plane) and the xoz-plane (H-plane). (**a**) At 35 GHz. (**b**) At 45 GHz.

A performance comparison of our work and some up-to-date MMW planar dipole antennas are summarized in Table 2. It can be observed that the proposed folded dipole has good balance for cross-polarization and the bandwidth. The proposed SIW-fed folded

dipole antenna has the widest impedance bandwidth while keeping a compact structure and a high radiation efficiency. Moreover, although a thick substrate with the thickness of 0.508 mm is used in this design, the proposed antenna still achieves a better cross-polarization level owing to the use of a folded structure.

**Table 2.** Comparison among planar millimeter-wave dipole antennas.

| Ref. | [8] | [9] | [11] | [12] * | [22] * | [23] | This Work |
|---|---|---|---|---|---|---|---|
| Center freq. (GHz) | 45 | 45 | 40 | 60 | 28 | 40 | 42 |
| Fabrication process | Single layer PCB | Single layer PCB | Single layer PCB | Single layer PCB | Two layer PCBs | Two layer PCBs | Single layer PCB |
| Imp. BW (%) | 13.6 (42.3–48.4 GHz) | 31.1 (36–50 GHz) | 36.2 (26.5–38.2 GHz) | 30 (50–68 GHz) | 17.5 (26–31 GHz) | 34.4 (33.5–47.4 GHz) | 55.7 (33.3–53.7 GHz) |
| $\lvert S_{11}\rvert$ (center freq.) | −20 dB | −18 dB | −15 dB | −19 dB | −25 dB | −11 dB | −25 dB |
| Gain (dBi) | 3.7–5.2 | 2.6–3.3 | 4.5–5.8 | 10.5–11.7 | 9.1 | 4.5–7 | 5.1–6.4 |
| $\varepsilon_r$ / $\tan\delta$ | 2.2/0.0009 | 2.2/0.0009 | 2.2/0.0009 | 3.38/0.0027 | 2.2/0.0009 | 2.2/0.0009 | 2.2/0.0009 |
| Feeding network | SIW | SIW | Microstrip | Microstrip | SICL | SIDL | SIW |
| Cross polarization level (dB) | −7 dB | −25 dB | −15 dB | −15 dB | −20 dB | −10 dB | −15 dB |
| Size (mm$^3$) | 6 × 26 × 0.508 | 5.5 × 38 × 0.25 | 10 × 30 × 0.25 | 9.2 × 10 × 0.2 | N. × N. × 0.608 | 13 × 20 × 0.608 | 11 × 14 × 0.508 |
| Max. Rad. Eff. | 82% | N.A. | 93% (Sim.) | N.A. | N.A. | N.A. | 80–95 |

* are designed with a 1 × 4 array, others are designed with one antenna element.

## 4. Conclusions

A SIW-fed printed folded dipole antenna is proposed for the MMW application. By cutting two gaps on the folded dipole, an additional resonance is generated, thus the operating bandwidth and impedance matching is greatly improved. Moreover, by appropriately adjusting the length of the DSPSL, the electric fields generated by the aperture and the connecting metallic vias are designed with an out-of-phase potential; thus the cross-polarization level of the proposed folded dipole is improved. The cross-polarization levels of the proposed antenna are approximately 9 dB lower than that of the conventional SIW-fed dipole antenna. The folded dipole antenna obtains a working bandwidth of 58.5% from 30.3 to 53.7 GHz and a very compact size. It can be used as the terminal antenna of the MMW communication system.

**Author Contributions:** Conceptualization, L.X. and K.F.; data curation, L.X.; formal analysis, Q.T.; methodology, L.X.; software, L.X. and K.C.; validation, K.F., L.X. and K.C.; investigation, L.X.; resources, K.F. and Q.T.; writing —original draft preparation, L.X. and K.C.; writing—review and editing, K.F. and Q.T.; visualization, Q.T.; supervision, K.F.; project administration, K.F.; funding acquisition, K.F. All authors have read and agreed to the published version of the manuscript.

**Funding:** This research received no external funding.

**Institutional Review Board Statement:** Not applicable.

**Informed Consent Statement:** Not applicable.

**Data Availability Statement:** Not applicable.

**Conflicts of Interest:** The authors declare no conflict of interest.

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
