# Peer review of "A Wideband Folded Dipole Antenna with an Improved Cross-Polarization Level for Millimeter-Wave Applications"

_applsci, doi:10.3390/app122111291_

Round 1
Reviewer 1 Report
A Wideband Folded Dipole Antenna with the improved Cross-2 polarization level for Millimeter-wave Applications is presented in the article. significant novelty couldn’t find in the article except the technique to improve cross-polarization level. However, a more detailed view of the antenna is proposed here with adequate theoretical analysis. The article can be improved by addressing the below comments.
1) The frequency span in the s-parameter picture (figure 5) should be increased to clearly understand the exact bandwidth of the proposed antenna.
2) it was mentioned that the simulation and measurement of Fig 9 agrees, however that is not the case. the discrepancy in the result should be justified.
3) The gain measurement is not done throughout the bandwidth, but it is limited until 50GHz, whereas the antenna bandwidth is until 55 GHz. This should be addressed in the text.
4) The curve style for cross-polarization is not legible.
5) The caption of the figure 4 can be improved.
6) Cross polarization improvement can be seen in many other articles. it is advised to mention how the mentioned technique is differ/improved from other published techniques.
Reviewer 2 Report
In the submitted letter, titled, “A Wideband Folded Dipole Antenna with the improved Cross-polarization level for Millimeter-wave Applications”, the authors proposed a low-profile planar millimeter-wave (MMW) folded dipole antenna fed by substrate integrated waveguide (SIW). It is basically a continued study of their previous publication to improve the cross polarization of the dual mode dipole antenna. The improved cross polarization is achieved by introducing the folded structure to the previously published printed dipole antenna. The results show improved cross polarization of the folded SIW fed dual mode antenna as compared to single mode folded SIW fed antenna. The gain of both antennas was comparable between 5 to 7dBi. Overall, the letter is very well written.
I have one comments:
1. In Fig. 8, the results for the antenna A, as the length Lst of the DSPSL increases, the amplitude of Gain(φ) is reduced, while the amplitude of Gain(θ) only has a small change. However, this trend is not seen for Antenna B. In fact, for antenna B, as the Lst length is increased, from 1.2mm to 1.6mm, the amplitude is reduced but from 1.6mm to 2mm, it increased again. Can authors please explain in detail about this observation?

Reviewer 3 Report
The manuscript entitled: "A Wideband Folded Dipole Antenna with the improved Cross-polarization level for Millimeter-wave Applications" is relevant to the Applied Sciences journal. The article is based on original experimental research. Overall, the paper is well prepared. Nevertheless, the article required some small changes:
- Line 72 – The ending of the sentence "... is shown in Fig. 1" is unnecessary because what is in this figure is already explained.
- Line 116 – What is the |S11| parameter? There was nothing about it before?
- Line 177 – We write Figure 6 and not Fig.
- Line 198 – „…simulated results ARE compared.”, but not IS.
- Page 7 – Since the results start from this page, what were the previous results and graphs about? This is where the discussion was held, and this is not here. Perhaps it should be otherwise. There should be a discussion of your results in relation to the work of others.
- References - Please check and prepare references following the template appropriate for the journal, available at:
https://www.mdpi.com/journal/applsci/instructions
Reviewer 4 Report
The proposal is excellent with significant scientific contributions. However, the authors should create a dedicated section discuss challenges of the approach and suggest future research direction or merged it with conclusion sections. The latest reference in the communication is 2018, it seems like the communication has been going round. What in the research area in 2019, 2020, 2021 and 2022 good four years update is missing in the communication. provide updated literature to prove that what you have done has not been done within these years.
Reviewer 5 Report
The authors have designed and fabricated a wideband folded dipole antenna with the improved cross-polarization level for millimeter-wave applications. The paper is well written and measurement results verify the simulated results. However, some modifications are needed in the manuscript, which should be considered as follows:
- In the abstract it is stated that “Compared with the up-to-date planar dipole antenna, the proposed folded dipole achieves the widest working bandwidth and lowest cross-polarization level.” However, there are not any comparison in the results section. It is suggested to add a comparison table to compare the bandwidth or other significant results in the manuscript with other antennas.
- The working bandwidth of the planar folded dipole is calculated as 58.5% (from 30.3 to 53.7 GHz). How it is calculated; if it is calculated such as 100*(53.7-30.3)/((53.7+30.3)/2), it should be recalculated and corrected in the manuscript.
- In “Figure 10. (a) Measured and simulated gain, cross-polarization and (b) efficiency of the proposed antenna.”, the part (a) and part (b) figures are identical and (b) efficiency of the proposed antenna is not shown.
- Kindly provide the overall size of the proposed antenna.
Round 2
Reviewer 5 Report
The authors have addressed all of my comments in the manuscript and the paper can now be accepted after considering some minor modifications:
- Add S11 parameter in the comparison table.
- The references should be in the numerical order.
- Follow the journal format for the manuscript.
